# Ambient Air Pollution and Sudden Infant Death Syndrome in Korea: A Time-Stratified Case-Crossover Study

**DOI:** 10.3390/ijerph16183273

**Published:** 2019-09-06

**Authors:** Myung-Jae Hwang, Hae-Kwan Cheong, Jong-Hun Kim

**Affiliations:** Department of Social and Preventive Medicine, Sungkyunkwan University School of Medicine, 2066 Seobu-ro, Jangan-gu, Suwon, Gyeonggi-do 16419, Korea (M.-J.H.) (J.-H.K.)

**Keywords:** air pollution, sudden infant death syndrome, case-crossover study, epidemiology, risk factor

## Abstract

Sudden infant death syndrome (SIDS) is an occasional cause of unexpected mortality in infancy. While various etiological factors have been hypothesized, air pollution has been consistently presented as an environmental factor. In this study, we aimed to estimate the risk of SIDS in relation to exposure to air pollution and the effects of its modifying factors. A mortality dataset with supplementary infant mortality survey data from Statistics Korea was used and combined the concentration of ambient air pollution data from AirKorea based on the date of death and residential addresses of the SIDS cases. Odds ratios (ORs) were estimated according to birthweight, gestational age, maternal age, and infant age using a time-stratified case-crossover study design. The risk of exposure to particulate matter of less than 10 μm in diameter (PM_10_), nitrogen dioxide (NO_2_), carbon monoxide (CO), and sulfur dioxide was estimated. The number of deaths due to SIDS was 454 (253 males and 201 females). The OR per 27.8 µg/m^3^ increment of PM_10_ was 1.14 (95% confidence interval [CI]: 1.03–1.25) and that per 215.8 ppb of CO was 1.20 (95% CI: 1.03–1.40) in all infants. In females, an increase in NO_2_ and CO levels was associated with a higher risk of SIDS in low-birthweight and preterm infants. The OR per 15.7 ppb increment in NO_2_ was highest among preterm infants, with a value of 5.12 (95% CI: 1.27–20.63), and low-birthweight individuals, with a value of 4.11 (95% CI: 1.74–9.72), at a moving average of 0 to 3 days. In males, however, no significant association was found. In the present study, exposure to air pollution was associated with an increased risk of SIDS. This association was more evident in susceptible infants with a low-birthweight or in cases of preterm birth.

## 1. Introduction

Over the past decades, there have been studies on the relationship between air pollution and daily mortality [1,2,3,4]. The health effect of air pollution is different across the population and vulnerable groups, including the elderly population, pregnant women, infants and children in early childhood, and those with chronic diseases. Reports on the vulnerability of infants are relatively scarce [5,6]. Sudden infant death syndrome (SIDS) represents the sudden unexpected, unexplained death of an infant without a demonstrable cause of death, apparently occurring during sleep [7]. Numerous studies have suggested risk factors for SIDS, such as apnea, cardiac arrhythmias, infective or inflammatory airway obstruction, rebreathing of expired gases, and thermal stress [8,9,10].

Recently, studies on SIDS in relation to ambient air pollution have been actively conducted [11,12,13,14,15], and potential exposure during the prenatal period [16,17] and direct exposures such as parental smoking after birth [18,19,20,21] have been reported to increase the associated risk. A possible explanation for the association between atmospheric air pollutants and SIDS is that infants within one year of age are rapidly developing, and their vulnerability varies greatly with effect modifiers in the short-term exposure to ambient air pollution, which even leads to death. Low-birthweight or preterm infants may have immature development of the brain compared to normal infants, reducing the ventilatory capacity for hypoxia [22] and inhibiting the respiratory control mechanism [23]. In addition, abnormalities of pulmonary development [24] and respiratory infections in neonates are promoted [25].

However, since daily deaths due to SIDS are very rare, the relevance of daily air pollution exposure using mortality data has not been well-defined due to the complexity of time-series modeling [26]. To minimize these biases, Litchfield et al. [27] assessed the risk of SIDS with daily exposure to ambient air pollution using a time-stratified case-crossover study in the UK. They showed a consistent association between an interquartile range increment of particulate matter less than 10 µm in diameter (PM_10_) and nitrogen dioxide (NO_2_), and SIDS. Short-term effects of air pollution on SIDS, including the lag effect, were estimated using this methodology. There is a further need to reconfirm these findings in another population and for the impact of effect modifiers to be estimated. In this study, we aimed to estimate the risk of SIDS association between potential short-term exposure to ambient air pollution using a time-stratified case-crossover study focused on effect modifiers from the national population.

## 2. Materials and Methods

### 2.1. Data on SIDS Mortality

We used the infant mortality data in the Republic of Korea (South Korea) provided by the Statistics Korea (Korean Statistical Information Service) MicroData Integrated Service [28]. This survey was based on the nationwide death and birth notifications of healthcare institutions, through which we obtained information on the pregnancy, birth, and delivery information of subjects who were linked to the population dynamics report on deaths less than one year after birth. This survey contains general information about the infant and mother of the infant, information on the death, and matters related to pregnancy and childbirth. The survey contents include the gender of the infant, date of death, year of birth, place of death, date and method of delivery, gestational age at birth, birthweight, maternal age, and status of birth of the infant. However, due to the self-report method employed to obtain information about the infant, there were missing values in supplementary items related to the mother and infant, such as educational level, smoking history of mothers, and marital status. SIDS cases were recruited from the year 2009 to 2013, based on the International Classification of Diseases’ 10th Revision (ICD-10) codes for the principal cause of death R95 that occurred within the territory of South Korea. The ages of all infants at death ranged between 0 and 365 days.

### 2.2. Ethical Statements

The study protocol was approved by the Institutional Review Board of Sungkyunkwan University (IRB #2018-03-026). No individual identifiers were provided in the dataset and informed consent was waived.

### 2.3. Exposure Data

South Korea is comprised of seven metropolitan cities and nine provinces. The 16 major regions consist of 253 health administrative districts; however, there are only 96 sites with air pollution monitoring stations in 16 regions, as of 2009. To ensure the absence of domestic air pollution monitoring stations and maximize the number of SIDS cases, we divided observations of air pollution of all monitoring stations from the Korea Ministry of Environment into 16 regions and calculated the daily average of PM_10_, NO_2_, carbon monoxide (CO), and sulfur dioxide (SO_2_). The measured data were obtained every hour and refined to calculate the daily average concentrations. We classified the monitoring station addresses of air pollution into major regions, according to the administrative district, considering that the infant addresses of the mortality data are provided based on 16 major regions. We combined ambient air pollution data and a mortality database on the infant residential address and date of infant death. We used the meteorological indexes provided by the Korea Meteorological Administration to estimate the daily average temperature, maximum temperature, and relative humidity nationwide. These data were collected for only 45 sites of 253 health administrative districts. We categorized the data into 16 administrative divisions, calculated the daily average level of meteorological indexes, and matched the mortality data on the infant residential address and date of infant death.

### 2.4. Measurement of Variables

In this study, we stratified cases based on effect modifiers such as sex, birthweight, gestational age, maternal age, and infant age at death, identified as a risk factor for SIDS [29]. Birthweight was stratified into normal (>2500 g) and low-birthweight (≤2500 g); gestational age was stratified into near term (≥37 weeks) and preterm (<37 weeks); maternal age was stratified into <35 years old and pregnancy of an advanced maternal age (≥35 years old); and infant age was stratified into 1–2 months and 3–11 months. We matched the concentration of daily average values as ambient air pollution and meteorological indexes according to the infant residential address and date of death. To evaluate the lag effect, we estimated the OR of SIDS per interquartile range (IQR) increment in PM_10_, NO_2_, CO, and SO_2_ from lag0 (same day as death) to lag3 and for a moving average of 0 to 3 days before the death.

### 2.5. Statistical Analysis

We conducted a time-stratified case-crossover study, which is suitable for estimating short-term exposures with consequences such as acute death, to evaluate the risk of SIDS in association with ambient air pollution [30]. This method has also been applied in previous studies on SIDS [31,32,33,34,35] and is compared to a set of ‘control’ times, where the exposure prior to the event represents the expected exposure distribution for non-event follow-up times [30]. Therefore, the greatest advantage of this method is that it can be applied only with a case group, without the need to select a control group. It can serve both as a case group and control group, so the number of samples can be as much as half of that of a general case-control study. Additionally, this method is primarily helpful to control the confounders caused by subject-specific factors over time, such as the sex of the subject. In order to take into account the need to control seasonal conditions and time trends, a control period was selected around two weeks before and after the death (1:4 matching). The results of case-crossover studies can be sensitive to the selection of the control period [35]. Therefore, we compared the effects of air pollution based on the application of two schemes, including bi-directional 1:2 (a control period around one week before and after the death) and 1:4 matching, and the different schemes showed similar results. We conducted a conditional logistic regression to estimate the risk of SIDS per IQR increment of air pollutants based on 1:4 matching bi-directional case-crossover (Figure 1).

Previous study has shown that the OR of SIDS has a significant non-linear relationship with the maximum daily temperature [33]. Based on this study, we identified the association between the daily maximum temperature and SIDS using the restricted cubic spline. We conducted the conditional logistic regression to calculate the OR per temperature increase (Appendix A), but the result did not show a significant relationship. Therefore, we performed conditional logistic regression stratified with effect modifiers, using a continuous measure of ambient air pollution, to obtain an effect estimate adjusted by the daily average temperature, maximum temperature, and relative humidity.

All statistical analyses were performed using SAS 9.4 (SAS Institute, Cary, NC, USA), and R version 3.5.3 (The R Foundation for Statistical Computing, Vienna, Austria). We calculated the OR and corresponding 95% confidence interval (95% CI) for each estimate.

## 3. Results

From 2009 to 2013, 454 (253 males and 201 females) infants died of SIDS nationwide. When stratified according to birthweight, gestational age, maternal age, and infant age, most male and female infants were included in the groups of a birthweight >2500 g, gestational age ≥37 weeks, and maternal age <35 years old. According to the age of survival of infants, about 39.2% were infants aged 1 to 2 months (Table 1).

Exposure levels of ambient air pollution and meteorological indexes of SIDS during the study period are shown in Table 2.

We conducted a conditional logistic regression to estimate the risk of SIDS per IQR increment of air pollutants based on 1:4 matching bi-directional case-crossover (Figure 1). The data showed consistent and stable results of the estimated effects of SIDS per IQR increment of air pollutants (Table 3).

The OR of SIDS per 27.8 µg/m^3^ increment of PM_10_ was 1.14 (95% CI: 1.03–1.25) at lag2 and the 215.8 ppb increment of CO was 1.20 (95% CI: 1.03–1.40) at lag1, which were statistically significant. The risk of SIDS per 15.7 ppb increment NO_2_ was highest, but not statistically significant. We stratified all infants by sex (Table 4).

The risk was higher in female infants, and the OR was increased 1.81 (95% CI: 1.28–2.54) times per 15.7 ppb increment in NO_2_ on the same day as death (lag0). For exposure to CO, this also significantly increased the risk of SIDS by 1.37 (95% CI: 1.07–1.76) per 215.8 ppb increment. We classified according to effect modifiers for all infants (Table 5), and the overall OR was consistently higher in low-birthweight and preterm infants at the moving average of 0 to 3 days.

The level of moving average of NO_2_ for 3 days was strongly associated with low-birthweight and preterm female infants. The OR per 15.7 ppb increment was 4.11 (95% CI: 1.74–9.72) in low-birthweight and 5.12 (95% CI: 1.27–20.63) in preterm birth. However, the maternal age of 35 years or older, which is defined as pregnancy of an advanced age, exhibited a different effect on modifiers according to sex. The OR per IQR increment of NO_2_ and CO in female infants born to mothers younger than 35 years was 1.93 (95% CI: 1.17–3.19) and 1.62 (95% CI: 1.13–2.33) at a moving average of 0 to 3 days, respectively. In male infants, there was a higher risk in the maternal age group of ≥35 years, but this was not statistically significant. Among 3–11 months female infants, the OR was 1.76 (95% CI: 1.00–3.11) per 15.7 ppb increment of NO_2_. We showed the lag effect of the IQR increment of air pollutants according to the effect modifier in female infants.

Figure 2a shows the effects of short-term exposure to air pollution by birthweight. In >2500 g infants, exposure to ambient PM_10_ also increased the risk of SIDS more at lag3 (1.21, 95% CI: 1.07–1.37); for exposure to NO_2_, the OR was the most increased to 1.62 (95% CI: 1.12–2.36) at lag0; and for CO, the value was 1.30 (95% CI: 1.01–1.66) at lag3. In ≤2500 g, except for PM_10_, all IQR increments of air pollutants were found to significantly increase the risk of SIDS on the same or previous day of death. Figure 2b shows a consistently high risk for preterm infants. The IQR increment of NO_2_ and CO showed the highest OR at a moving average of 0 to 3 days, and they also increased the risk at lag0 and lag1. In Figure 2c, the risk of exposure to ambient air pollutants born to mothers younger than 35 years was greater and statistically significant. In female infants aged 3–11 months, the risk of SIDS per IQR increment of NO_2_ and CO was increased. In particular, the OR per IQR increment of NO_2_ on the day of death showed a strong association, with a value of 2.28 (95% CI: 1.44–3.62) (Figure 2d). However, PM_10_ also showed an increased risk for infants aged 1–2 months.

## 4. Discussion

We estimated the risk of SIDS due to short-term exposure to air pollution using a nationwide infant mortality survey in South Korea by applying a time-stratified case-crossover study. We could clarify that short-term exposure to air pollution, including PM_10_, NO_2_, and CO, increases the risk of SIDS. When stratified by sex, female infants were more susceptible to the effects of air pollution, and factors such as low-birthweight, preterm delivery, younger maternal age, and infants’ postnatal age, were associated with a higher risk of SIDS after short-term exposure to air pollution. The OR in susceptible infants born with a low birthweight was higher than those born with a normal weight, as the lag effect was shorter. Moreover, the risk was higher for shorter lag periods. In female preterm infants, the OR was higher in infants with a gestational age ≥37 weeks and decreased as the lag increased.

In females, the risk of SIDS with higher environmental exposure to NO_2_ and CO has been reported to be an important risk factor for SIDS in previous studies [11,12]. In most studies, the proportion and risk of SIDS were higher in male than female infants [16,36,37]. These studies do not support the theory of the risk factors of male predominance in SIDS. The evidence for the risk factors for SIDS on sex is still unclear, and in this study, the risk effect in terms of exposure to air pollution was consistently higher in female infants. Although few studies could be used to determine that the risk effect of SIDS due to short-term exposure to ambient air pollution is different according to sex in South Korea, it can be estimated based on hypotheses of several biological mechanisms. Mechanisms supporting the differential effects of ambient air pollution by sex are uncertain, but it has been suggested that biological factors may have a stronger impact on females due to the lung volume, the responsiveness of hormonal effects, and the systemic regulation of chemical delivery [37].

We found that the risks of SIDS from short-term exposure to air pollution in low-birthweight and preterm infants were greater than in infants born with a normal birthweight or born in term. Knöbel et al. [38] reported an approximately threefold increase in the incidence of SIDS due to the exposure to higher average daily concentrations of fine particles in infants living in Taiwan, where the visibility was lowest because of air pollution. In addition, previous studies have reported that the shorter the gestational age, the higher the risk of neonatal death [39,40]. As shown in the results of this study, the risk of exposure to ambient air pollution in sensitive groups may be greater. Ritz et al. [12] showed a statistically significant increase in the risk of SIDS per increment of 1 ppm of CO of 1.46 (95% CI: 1.09–1.94) and a value of 1.26 (95% CI: 1.06–1.50) per 10 µg/m^3^ increment of PM_10_ in low-birthweight or premature infants. Additionally, Bobak et al. [41] reported that NO_2_ and SO_2_ increased the risk of SIDS in ≤2500 g infants in an ecological study, and Lipfert et al. [42] also showed that PM_10_ increases the mortality rate of SIDS to 3.48 per 1000 live births in born to low-birthweight post-neonatal.

Although older mothers are known to delay the development and growth of infants, the maternal age was not a risk effect modifier in this study, but the risk of SIDS was significantly higher in infants from mothers younger than 35 years, in contrast to previous studies. We also found that the risk of SIDS from exposure to air pollution was greater in 3 to 11 months-old infants than in 1 to 2-month-old infants. Previous studies have reported that as the infant develops, the temporary protective mechanism in the neonatal period disappears and the breastfeeding rate affects the susceptibility to SIDS [43,44,45]. Therefore, a combination of multiple factors in infants over 3 months of age may increase the risk of SIDS associated with exposure to air pollution. In addition, studies have reported on the relationship between an older maternal age at birth and infant mortality [46,47].

In this study, we evaluated the risk of SIDS in association with short-term exposure to air pollution. However, when considering the nature of exposure to air pollution, there were some limitations. Firstly, we matched exposure data based on pollutant exposure levels from administrative districts of 16 provincial and metropolitan levels of the infant’s residential address. Therefore, the spatial resolution of the air pollution status was lower, resulting in a lowering of the validity of the absolute level of air pollutants. However, day-to-day variation in exposure to air pollutants, which was our main determinant of exposure level in our study, may have been less affected by this processing. Actually, in order to prove this limitation, Kim et al. [48] analyzed the association with allergic diseases using diverse estimation methods for personal exposure to air pollution considering the subject’s residential environment and address. As a result, although the exposure resolution of air pollution was somewhat different, the risk effect was stable and consistent. Secondly, we did not consider the exposure from maternal smoking. Globally, for infants under one year of age, it is known to be a risk factor for SIDS. Particles of an atmospheric composition and tobacco smoke, such as CO, are similar and affect similar pathways in the respiratory system. Exposure to tobacco smoke in infants is associated with an increased risk of acute lower respiratory illnesses, including bronchiolitis, reduced pulmonary function, and asthma [20]. Thirdly, the diagnosis of SIDS was not based on the autopsy. According to a strong cultural antipathy against an autopsy, cause of death in this country is mostly based on a clinical basis, resulting in selection bias directed away from the null. Finally, due to the intrinsic limitations of questionnaire surveys, the risk effect could not be assessed based on maternal information. Every year, the supplementary infant mortality survey collects general information on the mother of the subject to obtain birth, pregnancy, and birth information for infant deaths of less than one year at domestic medical institutions and is conducted by Statistics Korea. However, since questionnaires were sent to the survey manager (medical records officer) by responders, maternal physical information, smoking status, marital status, and educational level were surveyed, but most subjects did not respond to the survey. Therefore, the information available in this study was limited. Most studies have reported on the risk factors of SIDS, mainly in terms of exposure to maternal smoking in pregnancy [21]. In addition, the proportion of non-responders for the level of education in the supplementary survey items of infant mortality was 27% and no income information could be obtained. Information regarding maternal socioeconomic status has been reported to have a great influence on infant mortality [49,50,51], but our study could not be stratified according to socioeconomic factors. Nonetheless, information on specific scientific factors involved in the risk of SIDS, such as birthweight, gestational age, maternal age, and infant age, was surveyed and available in this study.

Currently, studies on air pollution-related SIDS have focused on long-term exposure and there have been few studies that have analyzed the association between SIDS and ambient air pollution based on a time-stratified case-crossover design using infant mortality supplementary data. This study has an advantage in that it analyzed 80 to 90 nationwide cases of SIDS per year through a case-crossover study for minimizing bias [30,52]. This method is valid for assessing the risk of a rare acute outcome, such as SIDS association with short-term exposure, and has been applied in previous studies to estimate the risk [27,30]. Our study also has the advantage of using data from a mortality survey, including the information regarding birthweight, gestational age, maternal age, and infant age, in association with exposure to the ambient air pollution data. In South Korea, long-term exposure to ambient air pollution has led to the study of infant mortality and disease prevalence, but there have been few studies on acute death such as SIDS. Therefore, it is significant that this study assessed the risk of death from short-term exposure to air pollution.

Mortality is one of the main health outcomes of air pollution studies because it represents the most extreme health effect. Studies on the event of a death in susceptible populations, such as infants, may provide more contrast on the effect of air pollution on human health. Particularly in vulnerable groups, it has been suggested that ambient air quality should be improved, as infants are more sensitive and responsive to environmental factors. The causes of infant mortality suggest the necessity of expanding policies regarding maternal and infant health to prevent the future possibility of a decreasing population quality and increasing social burden.

## 5. Conclusions

This study suggested that the risk of acute infantile death, such as SIDS, might be increased in short-term exposure to air pollution. Among female infants, it was shown that they are susceptible and sensitive to air quality, especially in low-birthweight, preterm, younger maternal age, and post-neonatal age infants. Based on these studies, we should establish environmental health policies for the improvement of ambient air quality to protect infants and young children supported by scientific evidence.

## Figures and Tables

**Figure 1 ijerph-16-03273-f001:**
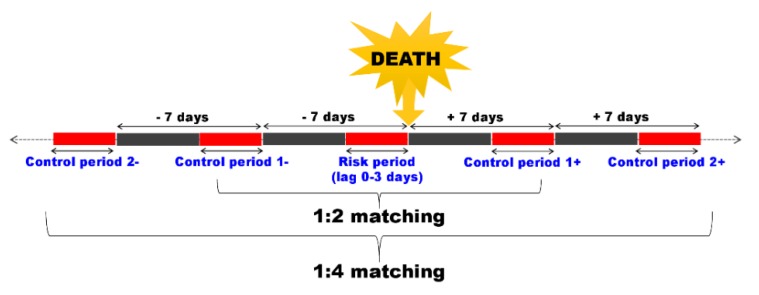
Schematic presentation of control selection in a time-stratified case-crossover study. Here, the case, a death (sudden infant death syndrome), was compared in terms of short-term exposure, such as air pollution, before and after the event.

**Figure 2 ijerph-16-03273-f002:**
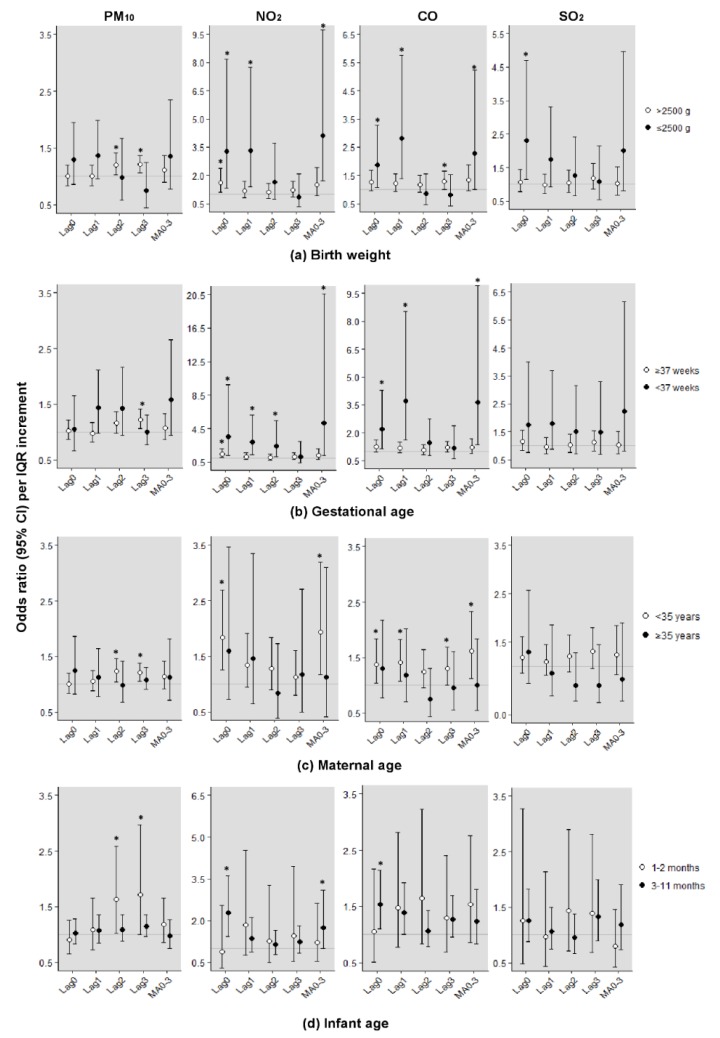
Odds ratios (95% confidence intervals) of sudden infant death syndrome (SIDS) per interquartile range increment in ambient air pollutants by effect modifiers in female infants. (**a**) Birth weight; (**b**) Gestational age; (**c**) Maternal age; (**d**) Infant age. * Statistically significant values (*p* < 0.05). Adjusting for daily average temperature, maximum temperature, and humidity. Estimated effects were expressed as the odds ratio per 27.8 µg/m^3^ increment in PM_10_ level, 15.7 ppb increment in NO_2_ level, 215.8 ppb increment in CO level, and 2.8 ppb increment in SO_2_ level. CI, confidence interval; PM_10_, particulate matter less than 10 µm in diameter; NO_2_, nitrogen dioxide; CO, carbon monoxide; SO_2_, sulfur dioxide.

**Table 1 ijerph-16-03273-t001:** General characteristics of the sudden infant death syndrome (SIDS) cases.

Variables	Total (*N* = 454)		Male (*N* = 253)		Female (*N* = 201)		*p*-Value ^a^
	*n*	%	*n*	%	*n*	%	
Birthweight							0.837
>2500 g	376	82.8	210	83.0	166	82.6	
≤2500 g	78	17.2	43	17.0	35	17.4	
Gestational age							0.092
≥37 weeks	377	83.0	207	81.8	170	84.6	
<37 weeks	77	17.0	46	18.2	31	15.4	
Maternal age							<0.0001 *
<35 years old	399	87.9	228	90.1	171	85.1	
≥35 years old	55	12.1	25	9.9	30	14.9	
Infant age							0.109
1–2 months	175	39.2	103	40.7	75	37.3	
3–11 months	276	60.8	150	59.3	126	62.7	

* Statistically significant values (*p* < 0.05). ^a^
*p*-Values were obtained by comparing the groups using the Chi-squared test or Fisher’s exact test.

**Table 2 ijerph-16-03273-t002:** Distribution of air pollutants and meteorological indexes.

	Mean	SD	Percentile					IQR
			Min	25th	50th	75th	Max	
Daily exposures								
PM_10_ (µg/m^3^)	49.6	28.3	8.9	31.9	43.5	59.7	275.9	27.8
NO_2_ (ppb)	24.6	11.5	3.3	15.6	22.1	31.3	76.3	15.7
CO (ppb)	529.5	214.1	188.9	385.7	478.2	601.6	1733.3	15.8
SO_2_ (ppb)	5.4	2.4	1.0	3.7	4.9	6.6	22.9	2.8
Temperature (°C)	11.8	10.2	−13.7	3.6	12.4	21.0	31.7	17.4
Relative humidity (%)	64.5	15.4	14.0	54.0	66.0	76.0	99.3	22.0

PM_10_, particulate matter less than 10 µm in diameter; NO_2_, nitrogen dioxide; ppb, parts per billion; CO, carbon monoxide; SO_2_, sulfur dioxide; SD, standard deviation; IQR, interquartile range.

**Table 3 ijerph-16-03273-t003:** Estimated effects in odds ratios of sudden infant death syndrome (SIDS) per interquartile range increment of ambient air pollutants in all subjects.

Lag (Days)	Odds Ratios (95% CI) ^†^			
	PM_10_	NO_2_	CO	SO_2_
Lag 0	0.99 (0.89–1.12)	1.19 (0.95–1.48)	1.10 (0.94–1.29)	1.09 (0.91–1.30)
Lag 1	1.05 (0.94–1.17)	1.15 (0.93–1.42)	1.20 (1.03–1.40) *	1.09 (0.92–1.30)
Lag 2	1.14 (1.03–1.25) *	1.11 (0.89–1.37)	1.13 (0.97–1.33)	1.03 (0.85–1.24)
Lag 3	1.10 (1.01–1.20) *	1.13 (0.91–1.40)	1.16 (1.00–1.35) *	1.02 (0.84–1.23)
MA0-3	1.09 (0.94–1.26)	1.16 (0.86–1.55)	1.21 (0.98–1.48)	1.08 (0.85–1.38)

* Statistically significant values (*p* < 0.05). ^†^ adjusting for daily average temperature, maximum temperature, and humidity. Estimated effects were expressed as the odds ratio per 27.8 µg/m^3^ increment in PM_10_ level, 15.7 ppb increment in NO_2_ level, 215.8 ppb increment in CO level, and 2.8 ppb increment in SO_2_ level. CI, confidence interval; PM_10_, particulate matter less than 10 µm in diameter; NO_2_, nitrogen dioxide; ppb, parts per billion; CO, carbon monoxide; SO_2_, sulfur dioxide; MA0-3, moving average of 0 to 3 days.

**Table 4 ijerph-16-03273-t004:** Estimated effects in odds ratios of sudden infant death syndrome (SIDS) per interquartile range increment of ambient air pollutants by sex.

Lag (days)	Odds Ratios (95% CI) ^†^							
	PM_10_		NO_2_		CO		SO_2_	
	M	F	M	F	M	F	M	F
Lag 0	0.95 (0.80–1.12)	1.04 (0.89–1.22)	0.85 (0.63–1.15)	1.81 (1.28–2.54) *	0.94 (0.76–1.16)	1.37 (1.07–1.76) *	1.02 (0.82–1.27)	1.22 (0.91–1.62)
Lag 1	1.06 (0.90–1.24)	1.05 (0.90–1.22)	1.01 (0.76–1.34)	1.34 (0.97–1.85)	1.11 (0.91–1.37)	1.33 (1.05–1.67) *	1.12 (0.89–1.40)	1.05 (0.81–1.37)
Lag 2	1.11 (0.98–1.27)	1.18 (1.01–1.37) *	1.04 (0.77–1.39)	1.18 (0.85–1.62)	1.16 (0.93–1.44)	1.10 (0.87–1.39)	0.98 (0.76–1.26)	1.08 (0.82–1.43)
Lag 3	1.00 (0.86–1.15)	1.17 (1.04–1.31) *	1.12 (0.84–1.50)	1.13 (0.82–1.56)	1.12 (0.92–1.37)	1.20 (0.96–1.51)	0.93 (0.73–1.19)	1.17 (0.88–1.56)
MA0–3	1.13 (0.92–1.39)	1.06 (0.86–1.29)	0.94 (0.63–1.40)	1.51 (0.96–2.38)	1.14 (0.86–1.49)	1.30 (0.95–1.78)	1.11 (0.81–1.51)	1.03 (0.71–1.49)

* Statistically significant values (*p* < 0.05). ^†^ adjusting for daily average temperature, maximum temperature, and humidity. Estimated effects were expressed as the odds ratio per 27.8 µg/m^3^ increment in PM_10_ level, 15.7 ppb increment in NO_2_ level, 215.8 ppb increment in CO level, and 2.8 ppb increment in SO_2_ level. CI, confidence interval; M, male; F, female; PM_10_, particulate matter less than 10 µm in diameter; NO_2_, nitrogen dioxide; ppb, parts per billion; CO, carbon monoxide; SO_2_, sulfur dioxide; MA0-3, moving average of 0 to 3 days.

**Table 5 ijerph-16-03273-t005:** Estimated effects in odds ratios of sudden infant death syndrome (SIDS) per interquartile range increment of ambient air pollutants at a moving average 0 to 3 days by susceptible groups.

	Odds Ratios (95% CI) ^†^											
	PM_10_			NO_2_			CO			SO_2_		
	T	M	F	T	M	F	T	M	F	T	M	F
Birthweight												
>2500 g	1.10 (0.94–1.28)	1.12 (0.88–1.43)	1.11 (0.90–1.37)	1.21 (0.88–1.68)	1.00 (0.64–1.56)	1.49 (0.92–2.42)	1.23 (0.98–1.54)	1.15 (0.85–1.57)	1.35 (0.96–1.88)	1.04 (0.80–1.36)	1.08 (0.75–1.55)	1.02 (0.68–1.52)
≤2500 g	1.20 (0.86–1.65)	1.17 (0.77–1.78)	1.35 (0.77–2.35)	1.39 (0.73–2.65)	0.76 (0.31–1.85)	4.11 (1.74–9.72) *	1.33 (0.84–2.10)	1.06 (0.58–1.94)	2.29 (1.00–5.23) *	1.37 (0.88–2.13)	1.18 (0.64–2.16)	2.00 (0.80–4.97)
Gestational age												
≥37 weeks	1.08 (0.93–1.26)	1.11 (0.88–1.40)	1.08 (0.87–1.33)	1.14 (0.83–1.58)	1.02 (0.65–1.58)	1.28 (0.79–2.08)	1.18 (0.94–1.47)	1.14 (0.84–1.55)	1.22 (0.88–1.70)	1.08 (0.83–1.40)	1.12 (0.78–1.61)	1.03 (0.70–1.51)
<37 weeks	1.33 (0.94–1.89)	1.27 (0.75–2.13)	1.58 (0.94–2.66)	1.86 (0.96–3.61)	0.75 (0.29–1.96)	5.12 (1.27–20.63) *	1.70 (1.03–2.81) *	1.18 (0.61–2.26)	3.65 (1.35–9.90) *	1.28 (0.75–2.16)	1.04 (0.54–1.99)	2.23 (0.81–6.15)
Maternal age												
<35 years old	1.12 (0.96–1.30)	1.14 (0.91–1.41)	1.14 (0.92–1.41)	1.22 (0.89–1.68)	0.89 (0.59–1.35)	1.93 (1.17–3.19) *	1.28 (1.02–1.59) *	1.11 (0.83–1.49)	1.62 (1.13–2.33) *	1.09 (0.84–1.42)	1.00 (0.70–1.43)	1.24 (0.83–1.84)
≥35 years old	1.13 (0.77–1.64)	1.14 (0.54–2.39)	1.13 (0.71–1.81)	1.39 (0.64–3.02)	1.66 (0.43–6.43)	1.13 (0.41–3.09)	1.18 (0.72–1.93)	1.48 (0.60–3.63)	1.00 (0.54–1.84)	1.28 (0.74–2.21)	1.71 (0.84–3.50)	0.73 (0.28–1.90)
Infant age												
1–2 months	1.18 (0.95–1.46)	1.16 (0.86–1.57)	1.19 (0.86–1.65)	1.01 (0.63–1.63)	0.91 (0.49–1.69)	1.21 (0.55–2.63)	1.30 (0.93–1.82)	1.20 (0.78–1.83)	1.54 (0.86–2.75)	0.97 (0.67–1.40)	1.03 (0.65–1.63)	0.80 (0.43–1.90)
3–11 months	1.03 (0.85–1.25)	1.11 (0.82–1.48)	0.98 (0.75–1.27)	1.27 (0.87–1.85)	0.98 (0.58–1.64)	1.76 (1.00–3.11) *	1.16 (0.90–1.51)	1.11 (0.77–1.60)	1.23 (0.84–1.80)	1.18 (0.85–1.62)	1.17 (0.76–1.80)	1.19 (0.74–1.91)

* Statistically significant values (*p* < 0.05). ^†^ adjusting for daily average temperature, maximum temperature, and humidity. Estimated effects were expressed as the odds ratio per 27.8 µg/m^3^ increment in PM_10_ level, 15.7 ppb increment in NO_2_ level, 215.8 ppb increment in CO level, and 2.8 ppb increment in SO_2_ level. CI, confidence interval; T, total; M, male; F, female; PM_10_, particulate matter less than 10 µm in diameter; NO_2_, nitrogen dioxide; ppb, parts per billion; CO, carbon monoxide; SO_2_, sulfur dioxide.

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
