# Peer review of "Ambient Air Pollution and Sudden Infant Death Syndrome in Korea: A Time-Stratified Case-Crossover Study"

_ijerph, 2019, doi:10.3390/ijerph16183273_

Round 1

Reviewer 1 Report

The manuscript is clear and in general well structured, and recommended to publish in the journal. For a better article, the reviewer suggests to add the study limitation including inaccurate air pollution data to estimate personal exposure because the ambient monitoring data used for this study is not very ideal in personal exposure to infant or maternity. Other than this, the content of the paper is within the scope of the journal and the manuscript is recommended to be acceptance.

Reviewer 2 Report

The authors attribute the ambient contamination to a wide zone and do not interpolate to the actual mother address.  Even that they are using a sound method some author argue  that air pollutions can be aborted and produce a delay effect not always an instant or immediate effect. Finally, I am concern on the pollutans´s spikes  that that can be produced briefly during a day, than can produce a physiologic effect,  but are compressed in a daily mean value.

But after point out this limitations, I think the article is solid, well written and usefull for the scientific community and other readers.

JSG

This manuscript is a resubmission of an earlier submission. The following is a list of the peer review reports and author responses from that submission.

Round 1

Reviewer 1 Report

In general the study is well designed and the findings are consistent with previous publications. However, there are some major defects in data analysis, which need the authors to address in major revision.

1. Lines 68-69, the authors said “However, due to the self-report method employed, there were a lot of missing values in matters related to pregnancy and childbirth.” I wonder whether the deaths with missing values were excluded for the analysis, or the missingness might bring any bias to the analysis. I want to see some analysis (at least percentage) for the missing.

2. Line 93-94, should “we stratified” be “we grouped”? “Stratified” is usually used for data analysis. Did the authors include sex, birthweight et al. in the analysis? How did they adjust for these “stratum” variables in their analysis?

3. Line 114, although conditional logistics regression is a conventional method for the case crossover design, has the authors considered conditional Poisson regression?

4. Did the authors include the lags one by one in the model or included them together? If later, how the authors handle the autocorrelation between the lagged air pollutants?

5. Line 158, how did the author calculate the “cumulative exposure effect of NO2”? The coefficient of the 3-day moving average cannot be deem a “cumulative effect” should be estimated using an additive cumulative time series model or at least a distributed lag model.

6. The evident nonlinear relationship between meteorological variables and deaths has been confirmed by many studies, however, I didn’t not see the authors consider this issue in their data analysis.

7. Harvest should be investigated and discussed in the study.

Some minor issues:

8. Line 83-86, should “measurement points” be “air pollution monitoring stations”? Do the authors means that they divide the monitoring station into 16 major regions according to their geographical location? There are “daily average” and “daily mean” in the sentences. What is the relationship between the two values? The data preparation was not described clearly here.

9. Line 113, please give more detailed description of 1:2 and 1:4 matching. Move Lines 134-139 and Figure 1 here.

10. Figure 2, odds rations should be plotted using logarithmic scale.

11. Line 178, what is the unit in “per unit increment in ambient air pollutants”?

12. Line 184, 194, Figure 1 should be Figure 2.

Author Response

In general the study is well designed and the findings are consistent with previous publications. However, there are some major defects in data analysis, which need the authors to address in major revision.

1. Lines 68-69, the authors said “However, due to the self-report method employed, there were a lot of missing values in matters related to pregnancy and childbirth.” I wonder whether the deaths with missing values were excluded for the analysis, or the missingness might bring any bias to the analysis. I want to see some analysis (at least percentage) for the missing.

A)     We meant that the number of death of data was complete, but that some of the information of the supplementary survey items on infant and mother were not available. Basic information including age, sex, birthweight, gestational age, and age of infant and mother were complete. Only some additional information such as educational level, smoking history of mothers, marital status had missing data. For example, the proportion of non-responders for the level of education was 27% (lines 314-316). We agree that the expression of the sentence was ambiguous in the context and revised it as follows. “However, due to the self-report method employed on information about the infant, there were a lot of missing values in supplementary items related to maternal and infant, such as educational level, smoking history of mothers, marital status.” (lines 75-77).

2. Line 93-94, should “we stratified” be “we grouped”? “Stratified” is usually used for data analysis. Did the authors include sex, birthweight et al. in the analysis? How did they adjust for these “stratum” variables in their analysis?

A)     In the case-crossover design, the control group is the patient (as a case) itself. For example, in the 1:4 matching case-crossover study, 1 is the original death event, and the 4 is self-control in the bidirectional control period. Therefore, the control group for the patient has the same value for the fixed factors such as sex, age, and birthweight, etc. Only the exposure level of the risk factors is different (Maclure et al., Ref 30). We compared exposure during the risk period on the day the infant (the case) died with the exposure during the control period before and after 7- and 14 days, respectively, as one set and assign a unique number. Then, a logistic regression analysis was conducted on each set and the point estimate for the entire set was obtained. Therefore, variables that do not change with time interval lapsed (e.g. sex, age, and birthweight) were not reflected in the model.

If we want to assess risk for these variables, we can perform stratified analysis. In this study, we tried to evaluate the risk of SIDS from exposure to air pollution according to these effect modifiers (sex, age, birthweight, and gestational age) as stratified variables.

3. Line 114, although conditional logistics regression is a conventional method for the case crossover design, has the authors considered conditional Poisson regression?

A)     Before analyzing this study, we identified the daily distribution of SIDS nationwide. Most SIDS in the same area nationwide on the same day were one death, and three were the maximum events (following figures 1 and 2).

<Figure 1>

<Figure 2>

The Poisson distribution is a discrete probability distribution of the number of events occurring in a given time period. In this study, the distribution of the number of death per day was observed, which did not fit the Poisson distribution. The number of daily SIDS was very rare, and most daily deaths occurred as 0 or 1, resulting in an extreme distribution. Therefore, we used the time-stratified case-crossover study to observe the effects of short-term exposure to air pollution. In this model, the dependent variable (event of SIDS) was a binary outcome, and we conducted the conditional logistic regression.

4. Did the authors include the lags one by one in the model or included them together? If later, how the authors handle the autocorrelation between the lagged air pollutants?

A)     We analyzed the effect estimate one by one per each lag from 0 to 3 days, then combined them in one figure. Lags were not included in one model.

5. Line 158, how did the author calculate the “cumulative exposure effect of NO2”? The coefficient of the 3-day moving average cannot be deem a “cumulative effect” should be estimated using an additive cumulative time series model or at least a distributed lag model.

A)     We obviously used the level of 3-day moving average of NO2. We calculated the average the level of lag 0 to lag 3 of NO2. It might have been confusing. We have revised them accordingly, from the cumulative exposure to the moving average (line 189).

6. The evident nonlinear relationship between meteorological variables and deaths has been confirmed by many studies, however, I didn’t not see the authors consider this issue in their data analysis.

A)     We enough considered what the reviewer’s comment. When divided into four seasons in Korea, the number of SIDS was 131 (28.9%) in spring, 88 (19.4%) in summer, 121 (26.7%) in autumn and 114 (25.1%) in winter from 2009 to 2013. When the daily average temperature was above 25 ° C, the number of SIDS was 40 (8.8%) of the 454 infants who died due to SIDS. Infant death due to SIDS did not occur frequently in the summer and when a daily average temperature such as a heat wave. In this study, SIDS events were very rare and did not show the nonlinear relationship between meteorological variables and deaths. This is the reason that we used the time-stratified case-crossover study.

7. Harvest should be investigated and discussed in the study.

A)     Previous studies have shown that there is a causal association between ambient air pollution and the incidence of daily morbidity and mortality. It has been suggested, however, that these events cannot be displaced by a long period of time. Rather, they are only being brought forward by few days to a few weeks in persons who are likely to be hospitalized or to die in a short period of time. However, it is not evident whether there are susceptibility to develop SIDS and whether who are susceptible and liable for the death. We did not assume harvesting effect on this study.

Some minor issues:

8. Line 83-86, should “measurement points” be “air pollution monitoring stations”? Do the authors means that they divide the monitoring station into 16 major regions according to their geographical location? There are “daily average” and “daily mean” in the sentences. What is the relationship between the two values? The data preparation was not described clearly here.

A1) First, we have classified the monitoring stations according to the administrative districts of 16 major regions of Korea based on the addresses of the infant’s residence. Air quality data of a specific day was calculated by averaging the measurements of all air quality monitoring posts in the same administrative districts. We revised the “measurement points” to “air pollution of all monitoring stations” in line 93.

A2) In response to the second comment, we have revised them to be consistent with the “daily average” (line 96, 117).

9. Line 113, please give more detailed description of 1:2 and 1:4 matching. Move Lines 134-139 and Figure 1 here.

A)     We moved figure 1 to line 137 and description was updated with detailed description of difference of the schemes. Requiring detailed results were moved to the supplement as a part of the sensitivity analysis (Supplement Table 1).

10. Figure 2, odds rations should be plotted using logarithmic scale.

A)     We are aware that the interval estimates of the odds ratios (ORs) are not symmetrically distributed. However, the OR itself is estimated as a logarithm of beta and it is customary not to apply a logarithm even in case the interval estimates are not symmetric.

11. Line 178, what is the unit in “per unit increment in ambient air pollutants”?

A)     We recalculated the odds ratio (OR) per interquartile range (IQR) increment of all air pollutants. Therefore, the IQR of air pollutants is presented in figure 2. Main text and abstract was revised accordingly.

12. Line 184, 194, Figure 1 should be Figure 2.

A)     As a reviewer’s comment, we revised the number of figures (lines 193-204).

Reviewer 2 Report

Manuscript ID: ijerph-548589

Title: Ambient air pollution and sudden infant death syndrome in Korea: a time-stratified case-crossover study

Authors: Myung-Jae Hwang, Hae-Kwan Cheong, and Jong-Hun Kim

This manuscript estimates the association between the daily averages of ambient air pollutants (PM10, NO2, CO, SO2) and the sudden infant death syndrome (SIDS) in Korea. They used governmental ambient air pollution monitoring data for this study, which is not very ideal in exposure as authors already noticed. The manuscript is clear and in general well structured, but some descriptions are somewhat inappropriate and unclear that need to be rephrased. The content of the paper is within the scope of the journal and the manuscript is recommended to be acceptance with minor revision. The below are some comments for a better paper.

Major comments:

-          Each pollutant uses a different OR increment arbitrarily, would suggest to use IQR increment for all pollutants.

-          Model need to control with other compounders such as season, pbl height, etc, if possible

-          Authors used air pollution levels, not air quality index (AQI) which is a different concept. Please use correct the word (eg., air pollution levels) instead AQI

-          Need to discuss more why infant gender cause a dramatic change. How other studies concluded by gender.

-          Describe how to match air pollution and infant’s residential address (eg., nearest data or spatial averages) and how about ozone data.

-          The results need to be compared with other studies’ results (eg., odd ratios or model estimates)

-    Conclusion: this manuscript use ambient air pollution data, not indoor or outdoor data. It should not be concluded to be related with indoor or outdoor air pollution exposure. 

Minor comments:

-      Line 50-51: rephrase the sentence because it should be better  

-      Line 98-99: rephrase to “we matched daily averages of ambient air pollutants and meteorological variables”

-      Line 108-109: rephrase this sentence and describe more details about the benefit of the model.

-      Line 112-114: Is this sensitive analysis? If so, clarify it. If not, please describe sensitive results.

-      Table 2: add max temperature

-      Lines 239-247: rephrase to “the exposure to ambient air pollution would reflect somewhat that to indoor air pollution. Ambient and outdoor are different. Rephrase all, It is difficult to understand what it meant.

-      Lines 245-247: delete the sentence. It would be not related with this manuscript

-      Lines 259-260: delete the sentence. It would be not related with this manuscript

Author Response

This manuscript estimates the association between the daily averages of ambient air pollutants (PM10, NO2, CO, SO2) and the sudden infant death syndrome (SIDS) in Korea. They used governmental ambient air pollution monitoring data for this study, which is not very ideal in exposure as authors already noticed. The manuscript is clear and in general well structured, but some descriptions are somewhat inappropriate and unclear that need to be rephrased. The content of the paper is within the scope of the journal and the manuscript is recommended to be acceptance with minor revision. The below are some comments for a better paper.

Major comments:

1. Each pollutant uses a different OR increment arbitrarily, would suggest to use IQR increment for all pollutants.

A)     Thank you for the suggestion. We revised all the results as the ORs per IQR increment of air pollutants across the manuscript.

2. Model need to control with other compounders such as season, pbl height, etc, if possible

A)     To take into account of the need to control seasonal conditions and time trends, we selected a control period around two weeks before and after the death (lines113-114). In the time-stratified case-crossover study, if we select the control period longer, it will cause seasonal fluctuations. Therefore, we considered this point and selected the control group around one and two weeks before and after death.

We estimated the effect of SIDS per interquartile range increment of air pollutants classified into four seasons in Korea (Table below). We could not find a significant effect of season on the SIDS risk, therefore, they were not presented in the manuscript.

<Table 1 inserted>

3. Authors used air pollution levels, not air quality index (AQI) which is a different concept. Please use correct the word (e.g., air pollution levels) instead AQI

A)     Following the reviewer’s comment, we revised ‘air quality index’ to ‘air pollution levels’. We rephrased it to ‘air pollution levels’ (lines 15, 93, 143).

4. Need to discuss more why infant gender cause a dramatic change. How other studies concluded by gender.

A)     Few epidemiologic studies have shown that the risk of SIDS due to short-term exposure to air pollution is different by sex. We have cited the reference for this scientific evidence based on several biological mechanisms (lines 247-252). We would like this point should be investigated in the forthcoming researchers on this subject.

5. Describe how to match air pollution and infant’s residential address (eg., nearest data or spatial averages) and how about ozone data.

A1) We presented more details about matching method in the lines 90-94 and added a reference (Kim et al., 2017) that can support the limitation of exposure resolution of ambient air pollution (lines 285-288).

A2) We matched the mortality data and daily 8-hr moving average of O3 in the same method as other air pollutants (PM10, NO2, CO and SO2) in this study. However, they did not show the statistically significant results and therefore, we did not present in this study.

<Table 2 inserted>

<Table 3 inserted>

6. The results need to be compared with other studies’ results (eg., odd ratios or model estimates)

A)     We added references to compare with other studies in lines 262-267.

7. Conclusion: this manuscript use ambient air pollution data, not indoor or outdoor data. It should not be concluded to be related with indoor or outdoor air pollution exposure. 

A)     Per reviewer’s comment, we suggested that policies should be established to improve air quality of the atmosphere, not indoor and outdoor air pollution (lines 344-345).

Minor comments:

8. Line 50-51: rephrase the sentence because it should be better  

A)     We wanted to mention that it is difficult to assess the risk based on short-term or daily exposure to air pollution on time-series modelling due to SIDS which is occurring 80 to 90 per year in South Korea. This is also the reason for the using case-crossover study in which patients themselves are used as control group. The confusion of the sentence was anticipated and it was rephrased to the following sentence. “However, since the daily deaths due to SIDS in South Korea are very rare, the relevance of daily air pollution exposure using mortality data is not well defined due to the complexity of time-series modeling.” (lines 56-58).

9. Line 98-99: rephrase to “we matched daily averages of ambient air pollutants and meteorological variables”

A)     As a reviewer’s comment, we corrected it to the following sentence. “We matched the concentration of daily average values as ambient air pollution and meteorological indexes according to infant residential address and date of death.” (lines 113-114).

10. Line 108-109: rephrase this sentence and describe more details about the benefit of the model.

A)     We rephrased the sentence and added details about the model (lines 124-128).

11. Line 112-114: Is this sensitive analysis? If so, clarify it. If not, please describe sensitive results.

A)     We conducted the sensitivity analysis. The similar results in different schemes are additionally presented in (Supplement Table 1).

12. Table 2: add max temperature

A)     We presented the level of percentile and maximum temperature in table 2. 

13. Lines 239-247: rephrase to “the exposure to ambient air pollution would reflect somewhat that to indoor air pollution. Ambient and outdoor are different. Rephrase all, It is difficult to understand what it meant.

A)     As a reviewer’s comment, we revised the contents of the entire paragraph because the paragraphs are inconsistent. However, we suggested that the maternal smoking and subsequent postnatal exposure are risk factors for infant mortality (lines 293-296).

14. Lines 245-247: delete the sentence. It would be not related with this manuscript

A)     We deleted this sentence.

15. Lines 259-260: delete the sentence. It would be not related with this manuscript

A)     We deleted this sentence.

Round 2

Reviewer 1 Report

I have to say I was not satisfied with the authors’ replies on my comments, especially for the comments 3, 5, 6, and 7. The conditional logistic regression is not the only way to handle the case-crossover data, and nonlinear relationship between deaths and meteorological variables should be taken into account in the analysis. I suggest the authors consult with a biostatistician on these questions. If the authors cannot address these question in their data analysis, at least they may address them in discussion.